# SARS-CoV-2 Neutralizing Antibodies in Chile after a Vaccination Campaign with Five Different Schemes

**DOI:** 10.3390/vaccines10071051

**Published:** 2022-06-30

**Authors:** Ximena Aguilera, Juan Hormazábal, Cecilia Vial, Lina Jimena Cortes, Claudia González, Paola Rubilar, Mauricio Apablaza, Muriel Ramírez-Santana, Gloria Icaza, Loreto Nuñez-Franz, Carla Castillo-Laborde, Carolina Ramírez-Riffo, Claudia Pérez, Rubén Quezada-Gate, Macarena Said, Pablo Vial

**Affiliations:** 1Centro de Epidemiología y Políticas de Salud, Facultad de Medicina Clínica Alemana Universidad del Desarrollo, Av. Plaza #680, San Carlos de Apoquindo, Las Condes, Santiago 7610658, Chile; claudiagonzalez@udd.cl (C.G.); paolarubilar@udd.cl (P.R.); carlacastillo@udd.cl (C.C.-L.); 2Instituto de Ciencias e Innovación en Medicina, Facultad de Medicina Clínica Alemana Universidad del Desarrollo, Av. Plaza #680, San Carlos de Apoquindo, Las Condes, Santiago 7610658, Chile; jhormazabal@udd.cl (J.H.); mcvial@udd.cl (C.V.); linacortes@udd.cl (L.J.C.); carolinaramirez@udd.cl (C.R.-R.); pvial@udd.cl (P.V.); 3Facultad de Gobierno, Universidad del Desarrollo, Av. Plaza #680, San Carlos de Apoquindo, Las Condes, Santiago 7610658, Chile; mapablaza@udd.cl; 4Public Health Department, Faculty of Medicine, Universidad Católica del Norte, Larrondo 1281, Coquimbo 1780000, Chile; mramirezs@ucn.cl (M.R.-S.); rquezada@ucn.cl (R.Q.-G.); 5Instituto de Matemáticas, Universidad de Talca, Avenida Uno Poniente #1141, Talca 3460000, Chile; gicaza@utalca.cl; 6Departamento de Salud Pública, Facultad de Ciencias de la Salud, Universidad de Talca, Avenida Uno Poniente #1141, Talca 3460000, Chile; lnunezf@utalca.cl (L.N.-F.); macarena.said@utalca.cl (M.S.); 7Escuela de Enfermería, Facultad de Medicina Clínica Alemana Universidad del Desarrollo, Av. Plaza #680, San Carlos de Apoquindo, Las Condes, Santiago 7610658, Chile; claudiaperez@udd.cl; 8Clínica Alemana de Santiago, Avenida Vitacura #5951, Vitacura, Santiago 7650568, Chile

**Keywords:** neutralizing antibodies, vaccines, BNT162b2, CoronaVac, AZD1222, COVID-19, SARS-CoV-2, cross-sectional design, vaccine-preventable diseases, viruses

## Abstract

Using levels of neutralizing antibodies (nAbs), we evaluate the successful Chilean SARS-CoV-2 vaccine campaign, which combines different vaccine technologies and heterologous boosters. From a population-based study performed in November 2021, we randomly selected 120 seropositive individuals, organized into six groups of positive samples (20 subjects each) according to natural infection history and the five most frequent vaccination schemes. We conclude that the booster dose, regardless of vaccine technology or natural infection, and mRNA vaccines significantly improve nAbs response.

## 1. Introduction

The SARS-CoV-2 pandemic has unprecedented challenges for its global, regional, and national control. The continuous emergence of the SARS-CoV-2 variants, jointly with the waning antibody titers from natural and vaccine-induced immunity, generates scenarios that maintain population susceptibility and risk of outbreaks [1,2]. Chile is not an exception, presenting one of the worst outbreaks in the world by mid-2020, but also with a globally successful vaccine campaign. The Chilean vaccination strategy combined different vaccine technologies (i.e., inactivated, viral vector and mRNA) and heterologous boosters [3].

The determination of nAbs is a well-known strategy for understanding the evolution of immunity against certain diseases. During the last two years, nAbs have been widely used to explore the progression of SARS-CoV-2. Recent studies have focused on comparing infection-induced, vaccine-induced, and hybrid immunity, and on analyzing the immune response to the SARS-CoV-2 variants of concern (VOC). This new evidence shows that vaccines and disease generate neutralizing antibodies, but nAbs are higher for individuals with heterologous and homologous vaccination schemes than those with natural immunity. Hybrid immunity protects better against severe outcomes of different VOCs. Additionally, nAbs titers varied significantly according to VOC and elapsed time from vaccination [4,5,6].

Despite the increasing literature covering this topic, only a few have analyzed inactivated vaccines. In this research, we aimed to compare the various Chilean vaccination schemes, using the presence of neutralizing antibodies (nAbs) as a correlate of immune protection against SARS-CoV-2 [7,8].

## 2. Materials and Methods

Serum neutralization capacity was measured using a pseudotyped vesicular stomatitis virus with a sequence encoding the enhanced green fluorescent protein as a reporter gene (VSV-GFP-Spike SARS-CoV-2 original Wuhan strain), kindly donated by Dr. Kartik Chandran [9]. Samples tested came from individuals enrolled in a population-based SARS-CoV-2 seroprevalence study performed by the same research team [10,11,12]. In November 2021, we collected 2198 serum samples from seven-year-old and older people, finding 97.3% of seropositivity. We used six groups of positive samples according to natural infection history and the five most frequent vaccination schemes, randomly selecting 20 individuals from each group (Table 1). Finance restrictions limited our ability to perform a higher number of nAbs analyses.

The amount of nAbs response was measured as the inhibitory concentration where 50% of the viral entrance is inhibited (IC50). IC50 was calculated for each serum by measuring the viral entrance of the VSV-GFP-Spike SARS-CoV-2 pseudotype capturing the amount of GFP fluorescence in each serum dilution. Briefly, serum serial dilutions from 1/50 to 1/51200 were incubated with VSV-GFP-Spike SARS-CoV-2 pseudovirus for 30 min, and then VEROE6 cells (ATCC) were infected with this virus. After 20 h, cells were washed, fixed in 4% paraformaldehyde, and GFP intensity was measured in a Cytation 3 (BioTeK). The resulting curve of each serum was analyzed through a dose-response nonlinear regression in Prism v9 Software (Graphpad) to calculate the IC50.

The statistical analysis considered the description of the median nAbs titers, including the 95% confidence interval for each category of analysis. We compared the median of nAbs titers after infection and after different vaccination schemes, using a Kruskal-Wallis test, with a *p*-value < 0.05. Additionally, we estimated the prevalence of positive nAbs response in each variable of interest, considering positive the subjects with IC50 in titers of 1/50 and over. Differences across subcategories were estimated using Chi2 statistical test or Fisher’s exact test. Data were analyzed using STATA statistical software (StataCorp. 2017. Stata Statistical Software: Release 15. College Station, TX, USA: StataCorp LLC.).

The Ethics Committees of the Universities el Desarrollo and Talca and the Facultad de Medicina of the Universidad Católica del Norte approved the study protocols. Informed consent was obtained from all subjects, if subjects were under 18, from a parent or legal guardian.

## 3. Results

We found nAbs response in 82.5% of the subjects, without significant differences by sex or age. The presence of nAbs is significantly higher in people with booster doses and non-smokers. Additionally, it varies according to vaccine platform used (inactivated, mRNA or viral vector recombinant) (Figure 1 and Table 2).

Figure 2 shows the level of neutralizing antibodies represented as median and interquartile values of the IC50 for each study group. In the left panel, when comparing nAbs levels, the group with only a basal immunization scheme has nAbs levels similar to those of the naturally infected patients (*p* value = 0.8425). In contrast, individuals who received a booster dose have a significantly higher level of nAbs compared to the other two groups.

On the right panel of Figure 2, analyzing the schemes by the different vaccines used, it is observed that the PPP scheme elicited the highest median nAbs response, without significant differences with the heterologous CCP scheme, but higher than the CCO scheme. On the other hand, all three booster schemes produced significantly higher nAbs levels than the natural infection group and the two basal schemes studied (CC and PP). Among the basal schemes, there are also significantly higher nAbs levels for the scheme with mRNA vaccines (PP) compared to inactivated vaccines (CC).

Figure 3 is a scatter plot showing the relationship between nAbs titers and time, using days since the last vaccine dose. It shows the waning of antibody titers for the groups with the basal vaccine scheme, but not for the groups with the booster doses, but also the follow-up was shorter for the latter groups.

## 4. Discussion

Our results demonstrate that vaccination with a booster dose significantly improves the neutralization of the virus, and this effect may be associated with the relatively lower impact of the circulation of the Delta variant observed in Chile compared to the previous SARS-CoV-2 variants in terms of cases, hospitalizations, and deaths [13]. By December 2021, 84.1% of the Chilean population had received a basal scheme vaccination and 56.1% a booster dose [13].

People with natural infection had a similar level of nAbs compared to people vaccinated with the basal schemes. However, nAbs levels in both groups, natural infection, and basal schemes, were significantly lower than those with booster doses, reinforcing the importance of universal vaccination, regardless of the history of the disease, as a strategy that confers higher protection.

Likewise, our results demonstrate the higher immunogenic potency of the mRNA vaccines, both in the basal and the booster dose schemes [8,14,15]. Other studies on healthcare workers from Chilean institutions support the higher neutralizing titers triggered by mRNA vaccines’ basal scheme [16]. A possible explanation might be the loss of antigenic sites in inactivated vaccines which only are exposed on a pre-fusion architectonic state of spike, which is necessary for infection dynamics [17].

Nevertheless, the heterologous booster scheme, combining inactivated and mRNA vaccines (CCP), displayed a heterogeneous response, including 15% of subjects without nAbs; this figure is zero in the other two booster schemes (CCO and PPP), and 10% in those with the PP basal scheme. Interestingly, the CCP group is younger than the CCO group (average 44 vs. 69 years old, respectively), but similar to the PPP group (average 44 years old), because the Chilean Health Authority restricted the use of ChAdOx1-S recombinant vaccine to people older than 55 years. A possible explanation for the proportion of non-responders with the CCP scheme may be the short time elapsed since the last vaccination. In fact, in two of the three subjects without nAbs, the sample collection was before 14 days, and this occurred in just one of the seventeen nAbs responders. The neutralization analysis detected significant differences according to vaccine technologies not seen in measuring total SARS-CoV-2 antibodies [11]. This added value, provided by neutralization studies, allows a deeper understanding of the antibody response to vaccines and natural infection to guide the public health response to the pandemic. Despite high vaccination coverage, we are still susceptible to new variants with the ability to evade the immune response, as was observed with the circulation of Omicron.

Finally, we found a lower nAbs response in smokers than non-smokers, consistent with studies suggesting a more inadequate humoral response in smokers [18].

The strength of this study includes the analysis of different vaccine technologies. In addition, it is a sample of subjects that comes from a population study and not from specific groups of the population. As for weaknesses, the moderate number of samples analyzed by vaccine technology does not include neutralizing antibodies against different variants of SARS-CoV-2, such as Delta and Omicron. Although previous studies have shown a correlation in neutralization for the different variants of SARS-CoV-2 [16], relevant changes have been detected for those with a greater capacity to evade the immune response. Finally, the short time elapsed since the application of the booster dose limited our ability to study nAbs half-life, further research is needed to address this matter.

## 5. Conclusions

We conclude that the booster dose significantly improves the levels of neutralizing antibodies against SARS-CoV-2, regardless of the vaccination scheme or the levels acquired by natural infection. Additionally, mRNA vaccine technology is strongly associated with higher neutralizing antibody levels than inactivated virus vaccines.

## Figures and Tables

**Figure 1 vaccines-10-01051-f001:**
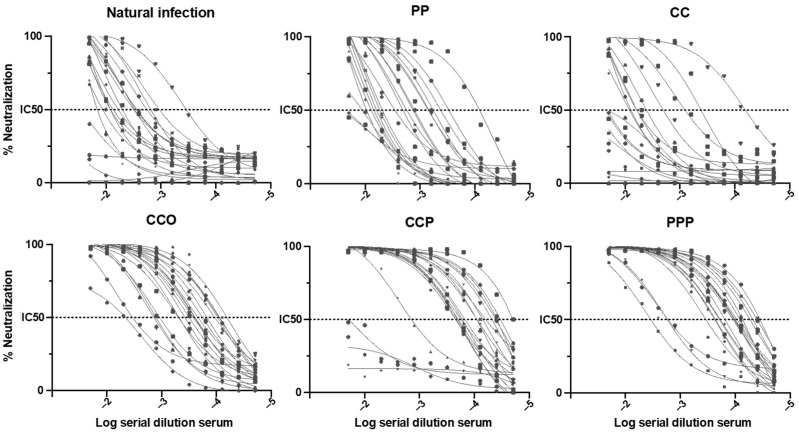
Neutralizing antibodies titration curves for the natural infection group and different vaccination schemes, Chile 2021. CC (CoronaVac CoronaVac), PP (Pfizer Pfizer), CCO (CC plus Oxford AstraZeneca), CCP (CC plus Pfizer), PPP (Triple Pfizer), with *n* = 20 for each group. The curves were fitted as a dose-response three parameters analysis. Each curve represent a patient, and the dots are the GFP fluorescence value for every serum dilution, using the GraphPad prism 9 software (Graphpad).

**Figure 2 vaccines-10-01051-f002:**
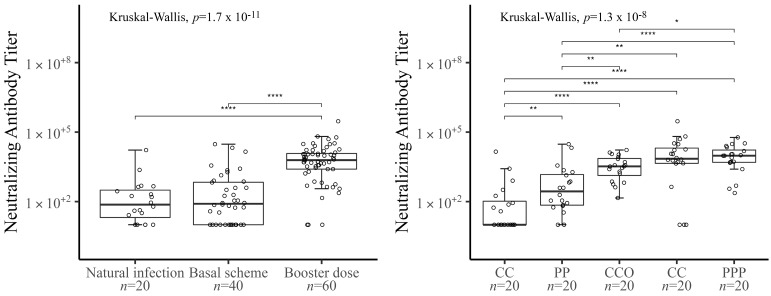
All samples were plotted as individuals points and graphed the median and interquartile for both panels. Left panel shows a comparison of nAbs titers between natural infection (*n* = 20), two dose schemes (*n* = 40), and booster dose (*n* = 60). The right panel shows a comparison of nAbs between the different vaccination schemes: CC (CoronaVac CoronaVac), PP (Pfizer Pfizer), CCO (CC plus Oxford AstraZeneca), CCP (CC plus Pfizer), PPP (Triple Pfizer), with *n* = 20 for each group. Symbols *, **, and **** denote statistically significant differences among the comparison groups; *p* ≤ 0.05, *p* ≤ 0.01 and *p* ≤ 0.0001, respectively. The statistical differences were performed with Kruskal-Wallis test, and a *p*-value < 0.05.

**Figure 3 vaccines-10-01051-f003:**
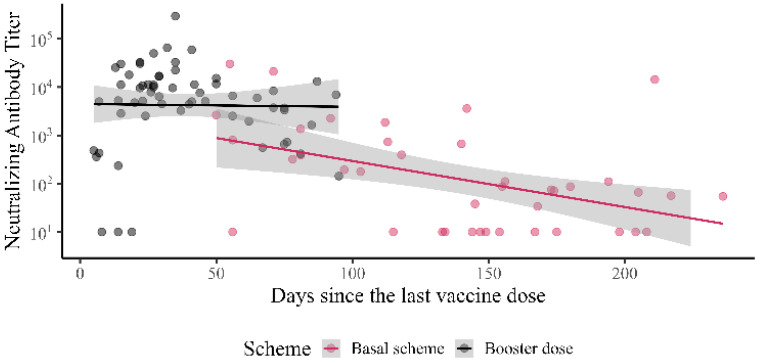
Scatter plot showing the relationship between nAbs titers and time, using days since the last vaccine dose.

**Table 1 vaccines-10-01051-t001:** Vaccination schemes and sample distribution, Chile 2021.

Vaccine Scheme Acronym	Description	N
CC (CoronaVac CoronaVac)	Basal scheme = two doses of Sinovac’s CoronaVac	20
PP (Pfizer Pfizer)	Basal scheme = two doses of BNT162b2	20
CCO (CC plus Oxford Astra-Zeneca)	Basal CC plus booster with ChAdOx1-S (heterolo-gous)	20
CCP (CC plus Pfizer)	Basal CC plus booster with BNT162b2 (heterologous)	20
PPP (Triple Pfizer)	Basal PP plus booster with BNT162b2	20
Natural infection	Non-vaccinated, but seropositive (Natural Infection)	20

**Table 2 vaccines-10-01051-t002:** Presence of Neutralizing Antibodies against SARS-CoV-2 among seropositive individuals according to selected variables, Chile November 2021.

Variable		*n*	Median nAbs (p25–p75) ^§^	Positive Ab Response	Prevalence (%)	*p*-Value
Total		120	730.7 (63.4–7757.1)	99	82.5%	
Sex	Male	43	485.4 (71.6–6939.6)	37	86.0%	0.31
	Female	77	1640.2 (42.3–8333.3)	62	80.5%	
Age group	7–19	16	336.1 (124.1–1084.1)	15	93.8%	0.20
	20–59	81	668.9 (38.1–11,076.7)	63	77.8%	
	60+	23	2637.1 (558–6939.6)	21	91.3%	
COVID-19 diagnosis	No	108	698.9 (55.5–7283.7)	87	80.6%	0.09
	Yes	12	2970.9 (141–12,809.6)	12	100.0%	
Presence of symptoms *	No	74	2293.3 (87.3–7627.8)	62	83.8%	0.41
	Yes	46	356.2 (40.4–9615.4)	37	80.4%	
Comorbidity ^†^	No	65	668.9 (60.8–6583.3)	54	83.1%	0.52
	Yes	55	732.6 (66–9578.5)	45	81.8%	
Tobacco	No	83	800.6 (86.8–5274.3)	73	88.0%	0.02
	Yes	37	558 (10–11,611.7)	26	70.3%	
Vaccine	No	20	75.4 (18–356.5)	15	75.0%	0.25
	At least one dose	100	2522.4 (98.8–9845.5)	84	84.0%	
Vaccine doses	Basal scheme	40	80.8 (10–700.7)	27	67.5%	0.00
	Booster	60	6172.3 (2522.4–12,346.1)	57	95.0%	
Vaccine scheme ^‡^	CC	20	10 (10–132.2)	9	45.0%	0.00
	PP	20	292.9 (68.8–1614)	18	90.0%	
	CCO	20	3305.1 (1184.5–7636.5)	20	100.0%	
	CCP	20	7105.5 (4456.6–24,035.9)	17	85.0%	
	PPP	20	9597 (4884.6–18,837.4)	20	100.0%	
	Natural infection	20	75.4 (18–356.5)	15	75.0%	

* COVID-19 compatible symptoms including fever, cough, odynophagia, dyspnea, headache, myalgia, chest pain, abdominal pain, diarrhea, fatigue, anosmia and dysgeusia. ^†^ Comorbidities including overweight and obesity, diabetes, high blood pressure, heart diseases, chronic respiratory diseases (asthma, COPD), cancer and hypothyroidism. ^‡^ Vaccine scheme acronym description in Table 1. ^§^ Interquartile range (p25–p75) = 25th and 75th percentile.

## Data Availability

Study dataset are available from the authors upon reasonable request.

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
