# Peer review of "SARS-CoV-2 Neutralizing Antibodies in Chile after a Vaccination Campaign with Five Different Schemes"

_vaccines, 2022, doi:10.3390/vaccines10071051_

Round 1

Reviewer 1 Report

The manuscript presents a cross sectional study to evaluate neutralizing antibody response in vaccinated patients in SARS-CoV-2 infection. The field of research is one of the hottest topics and for this reason I recommend the publication of the manuscript after revisions. I leave my comments to authors as suggestions to improve the manuscript:

-the introduction is to much general on the pandemic well-known data, in fact, it lacks on an update presentation of previous works evaluating neutralizing antibodies in population with combined vaccines.

-I suggest to insert a specific chapter explaining the detailed statistical analysis applied.

-I suggest to modify the manuscript including a figure showing at least some representative inhibition curves.

-In reference 2 journal information is lost

Reviewer 2 Report

This is an interesting manuscript and addresses a major issue in counteracting Covid-19 infections. It provides some useful comparisons on the efficacy of different vaccines, including the superiority of mRNA vaccines and the value of boosters.

However, as a retrospective analysis using randomly selected individuals to populate each group, I have some concerns.

The first is because only 20 individuals were selected in each group despite a much larger collection of samples being taken. What limited sample size? Resources available for analyses? Or the number of samples available for the smallest group? Some extra explanation is needed. 

The second question is about the lack of longer-term data on neutralizing antibodies following boosters. I understand that boosters had only been recently given and from this type of study it is not possible to follow up and get subsequent data, however, some sort of prediction of the neutralizing antibody half-life - or discussion about the time to fall to ineffective titres might be useful. 

The manuscript is clearly written, and the English is good. Indeed, as an English speaker I noted only a couple of places where a small improvement was needed. In lines 36-37, it might be better to write 'Chile is not an exception'. In line 67, it might be better to write 'If subjects were under 18". However, these are minor changes.

Author Response

Response to Reviewer 2 Comments

This is an interesting manuscript and addresses a major issue in counteracting Covid-19 infections. It provides some useful comparisons on the efficacy of different vaccines, including the superiority of mRNA vaccines and the value of boosters.

However, as a retrospective analysis using randomly selected individuals to populate each group, I have some concerns.

Point 1: The first is because only 20 individuals were selected in each group despite a much larger collection of samples being taken. What limited sample size? Resources available for analyses? Or the number of samples available for the smallest group? Some extra explanation is needed. 

Response 1: The availability of resources indeed limited the sample for nAbs. The samples were taken all at the end of 2021 in a cross-sectional seroprevalence study. We had funds to perform 120 neutralizing tests, which represented around 6% of the seropositive results. An allusive phrase was incorporated in the Materials and Methods section (Line 65): “Finance restrictions limited our ability to perform a higher number of nAbs analyses.”

Point 2: The second question is about the lack of longer-term data on neutralizing antibodies following boosters. I understand that boosters had only been recently given and from this type of study it is not possible to follow up and get subsequent data, however, some sort of prediction of the neutralizing antibody half-life - or discussion about the time to fall to ineffective titers might be useful. 

Response 2: This is a relevant question, that we as a team will try to address in the new round of seroprevalence, that is underway at this moment. We will have people with one and with two booster doses. We added a sentence acknowledging this limitation in our study (line 189): “Finally, the short time elapsed since the application of the booster dose limited our ability to study nAbs half-life, further research is needed to address this matter.”

Point 3: The manuscript is clearly written, and the English is good. Indeed, as an English speaker I noted only a couple of places where a small improvement was needed. In lines 36-37, it might be better to write 'Chile is not an exception'. In line 67, it might be better to write 'If subjects were under 18". However, these are minor changes.

Response 3:  We amended the text following your suggestions.

Reviewer 3 Report

Dear authors,

thank you very much for this well-written and interesting manuscript!

My only comment is on the inscription of Figure 1; this should be enlarged so that the p-values become readable in the printed version of the manuscript.

My recommendation to the Editor is to accept the manuscript in the present form, subject to the change in Figure 1.

Best wishes!

Author Response

Response to Reviewer 3 Comments

Dear authors,

thank you very much for this well-written and interesting manuscript!

Point 1:  My only comment is on the inscription of Figure 1; this should be enlarged so that the p-values become readable in the printed version of the manuscript.

My recommendation to the Editor is to accept the manuscript in the present form, subject to the change in Figure 1.

Best wishes!

Response 1: We thank the positive evaluation of the reviewer and the comment to improve the quality of the manuscript.  We enlarged the text in the figure, following your suggestion.